# (In)active God—Coping with Suffering and Pain from the Perspective of Christianity

Franjo Mijatović 

Faculty of Medicine, Social and Humanistic Sciences in Medicine, University of Rijeka, 51000 Rijeka, Croatia; franjo.mijatovic@medri.uniri.hr

**Abstract:** Colloquially, suffering and pain are usually and exclusively concerned with the human body. Pain and suffering are clearly objective facts, as well as lasting and memorable experiences. Are suffering and pain purely biological phenomena and neurological states, or can they be interpreted by culture, religion, philosophy, sociology, Christianity, etc.? To what extent can it, therefore, be said that the body is sufficiently cognitively, motorically, and sensibly equipped to accept or reject unpleasant situations. Except biological, neurological, and medical, i.e., physical, views about suffering and pain, the Christian solution is one of the essential elements of human life which can serve as a bridge between adaptive and cognitive management and control of the body and mind and learned (parents, culture, society) patterns of dealing with pain and suffering. Our article aims to show how Christianity, in describing suffering and pain as the physiological fact and subjective experience, can be gathered up into a meaningful whole and a powerful sense of *(in)active* God.

**Keywords:** pain; suffering; body; evolution; God; narration; action; Christianity

## 1. Introduction

Pain and suffering are complex and difficult questions, pondered by individuals and society at large. It does not concern only the past and present but the future as well. As these topics entail internal contemplation and theoretical exploration, they require a systematic examination. In the last few decades, there have been efforts in exploring and researching these topics from biological, evolutionary, cognitive, and medical perspectives. Yet, these individual efforts do not provide a holistic answer to such multiplex questions. While the contribution of these disciplines is necessary, of benefit would be to consider another point of view—a spiritual perspective. Therefore, this article will, in two parts, explore the scientific definitions of pain and suffering that do not instruct how one should deal with pain and suffering, therefore, distancing God from these experiences. Proponents of evolutionary theory, on the other hand, believe that suffering is simply a way "taken up into God's more comprehensive and gracious action in the created world" (Peters 2013, p. 115), while Deane-Drummond sees suffering as "the securing of some outweighing good" (Deane-Drummond 2008, p. 16). In any case, coping with suffering cannot be prescribed by biology or evolution, but will always require additional aids that are far beyond the conceptual framework of science.

The second part of the article will discuss pain and suffering through a human–divine (man–God) connection, and in this connection, there are easy, difficult, and formidable aspects. Pain and suffering belong to the difficult, if not impossible aspects. Suffering requires an internal effort, a transcending of oneself. A concrete answer to suffering cannot be given, as it is a part of life that does not have a roadmap. The difficulty lies in the fact that it is a personal state that everyone must endure individually, and therefore, the solution to suffering must be personal. There are no universal answers. One possible answer is in the Christian credo "memoria passionis, crucis, mortis et resurectionis Jesu Christi" (Metz 1978, p. 175). God's salvation does not suffer human limitations. Finally, the

bible tale of Job is based on historical experiences of faith, and it is a testimony of God's intervention in human history.

Therefore, this article will take a multidisciplinary approach to answer the conundrum of pain and suffering, from evolutionary, cognitive, psychological, and Christian points of view. More weight will be given to humanistic disciplines than to experimental approaches, as the former talks of pain that can be in touch with life. For suffering to be bearable, if not desirable, this formidable experience must be given a personal meaning. The following sections will look at how Christianity can provide an answer through meaning.

## 2. Pain and Suffering from the Perspective of Science

### 2.1. The Mystery of Pain and Suffering

The subject of pain has become an integral part of many sciences, primarily those sciences which were prefixed with "bio-". Thus, we have biology, biopolitics, bioethics, bioeconomics, biotechnology, biotheology, etc. Relying on the historical meaning of the concept of pain, it is observed that its understanding was burdened with shortcomings of definition and is subject to criticism and debate. The often cited and discussed definition of pain defines pain as "an unpleasant sensory and emotional experience associated with actual or potential tissue damage, or described in terms of such damage" (IASP 2020). Therefore, pain is not so difficult to define since, according to most prevailing definitions, it is related to the body, although the opposite definitions increasingly displace pain from the physical, i.e., bodily. However, we will agree that suffering is still different from pain, and it is related to human experience, and experiences as such are very different and not subject to scientific verification. Since the spirit of postmodernism seeks the equalization of everything or the conciliatory equal value of almost everything, then it also seems that this disregard for the difference between pain and suffering has gone in exactly that direction: on biological and physiological legality.

On the other hand, the modern scientific study of suffering is mixed with ambivalent feelings, and there is a kind of apprehension regarding the man who suffers and his socio-cultural order. Different perceptions of the study of suffering and its social consequences often determine the relationship between the spiritual and natural sciences. It seems that suffering, along with pain, is still a bridge between the spiritual and natural sciences. Pain and suffering, as objective facts for a long time, especially large-scale suffering (natural disasters), have often been the subject of philosophical, theological, and psychological sciences. On the other hand, philosophers, and especially theologians, believe that the suffering of Auschwitz, Aleppo, Srebrenica, etc., cannot even be explained otherwise, or it could be said that it cannot be understood otherwise rather than within the spiritual sciences.

Pain and suffering with the development of experimental sciences and pharmacology have become the centre of our interest and a kind of hope and consolation in their endurance, ultimately, a solution. In any case, especially today when we see great advances in technological medicine, we would like to avoid the shameful and severe consequences of pain, suffering, aging, etc. It has already been mentioned that in the opinion of both the public and the specialized (scientific), the body is the one that produces pain and suffering. In this sense, the physical body was often simply neglected: "It is not, of course, that modernist philosophy has shown any great interest in the organic substantial body as such, but rather, in the human as the abstract universal marker of the site of foundational voice, vision and vitality" (Shildrick 2002, p. 48). In this context, many theorists and scientists expect paradigm shifts in the natural science methodology as well so that the phenomenon of the body can be scientifically processed.

In addition to the general, almost everyday understanding of suffering within modern philosophies of spirit, scientific psychology, and cognitive sciences, suffering is observed primarily phenomenologically and defined synonymously, and therefore, its meaning is of limited use. For this purpose, the deictic examples are used from the I-perspective to describe the physical experience of pain: I am broken, my being is empty, something deep inside me torments me, I feel uncomfortable, I am in agony, I am afraid, I am lamentable

and miserable, etc. Therefore, E. Cassel's definition of suffering is often cited in support of the above description of phenomenal suffering as "the state of severe distress associated with events that threaten the intactness of the person" (Cassel 2004, p. 32). An essential feature of suffering according to Cassel is reflected in subjectivity. The suffering of one person would be in a sense the private suffering of that person, that is, the person has a certain relationship to her suffering and her feeling of suffering that is fundamentally different from the relationship of another person to her suffering, i.e., the difference is manifested in *how to be* such a sufferer. Thus, the aforementioned subjective experience of suffering is often limited to sensory experience. Moreover, a whole spectrum from simple to complicated experiences of suffering emerges. According to Svenaeus, suffering is "the way the whole world appears to us, opening up the world to the person in a certain tone or colour" (Svenaeus 2014, p. 409). Drew Leder (1990), in his very influential book, *The Absent Body*, talks about the body and then pain from a third-person perspective, while most phenomenologists believe that suffering is related to a person, her experiences, and the experience of suffering. Suffering in the sense of meta-suffering signifies the ability to be able to think about one's own suffering states and is closely related to various theories of suffering, by the ability to think about one's own experience, happenings, and coping with suffering that one can not only apply to oneself but can also influence others. Human suffering is also marked by linguistic reflection and introspective communication (Wierzbicka 2014), and in this sense, philosophers and scientists speak of suffering as an ability possessed by a man in a special way. The suffering that concerns me or that threatens me is an ever-changing state of my consciousness, my being, and my experience. The notion of suffering, on the other hand, has a long tradition in Christianity, and it has always been subject to narration and the transcendence of oneself as a sufferer. In evolutionary research, pain and suffering are too often separated into two separate entities. In the constructivist sense, today, there is more and more talk about the unity of pain and suffering, about the annulment of the differences between these two entities (Duffee 2019).

After a sketchy theoretical presentation of the concept of pain and suffering in contemporary discussions, this paper does not intend to enter furthermore into their specialized meanings in certain sciences. However, it can be said that an important characteristic of the phenomenon of suffering is in its subjective quality (first-person perspective), whereas pain can be more or less described as an objective phenomenon (third-person perspective). Many humanities scientists have over-ontologized and stretched the notion of suffering too much, trying to encompass under this notion all human experiences that most *threaten the intactness of the person*. This subjective area of suffering experience is then often confronted with the objective area of physical pain. The question is how can there be a subjective experience in a biological universe since the *coexistence* of pain and suffering is literally mysterious? The concept of phenomenal suffering as an ontologizing area of subjective experience leads to one form of the modern variant of Cartesian dualism (Bueno-Gómez 2017). However, there is no denying the existence of suffering experiences, although these subjective experiences are described in everyday speech by psychological predicates (I am anxious, depressed, I cannot carry on, I feel miserable, etc.). What remains problematic is how suffering conditions can be defined objectively. Do other living beings suffer? What about insects? Does a fly suffer when children tear off its legs? And what about even more primitive creatures such as amoebae? Considering other beings and respecting their different intuitions, it seems at first glance that visible actions alone do not give any unambiguous conclusions about the suffering of other living beings. However, how do we know that man suffers and fish do not? Is it possible in principle to imagine a *humanoid robot* that perfectly mimics human actions without any sense of suffering? With the triumph of modern natural sciences and the associated empiricist-materialist-shaped worldview, these and similar issues become central.

### 2.2. Empirical Research on Pain and Suffering

Evolution suggests to us with its empirical research that all biological processes, including primary pain, and then suffering as its *intangible* consequence, can be explained by body functions and neural networks. For the biologist, the body is a biological, relative concept based on material chemical substances, its experiences, and the environment that offers stimulus. So Lynne U. Sneddon writes: "The definition of human pain suggests that there are two components: firstly, a stimulus that could or does cause damage is perceived (termed nociception) and secondly, this leads to a psychological state where an individual experiences suffering or discomfort (termed pain) [ . . . ] injured squid fled from predators at a greater distance than non-injured squid; thus, the response to tissue damage has evolved as a survival tactic." (Sneddon 2019, pp. 1, 4) What does empirical research tell us primarily about pain? If the brain of a rainbow trout that possess nociceptors, which are similar to those in mammals, is electrophysiologically stimulated, there are changes associated with pain in physiology and behavior and greater brain activity. For the body to respond properly to a harmful event, the sensory system helps to *detect* tissue damage. In this way, animals that respond to tissue damage are more likely to survive and reproduce than those that do not have the ability to detect damage. Aversion to something in animals can be recognized through elevated heart rate or elevated levels of stress hormones. From any change in behavior, it can be concluded that a particular organism experiences injury or pain. The electrophysiological properties of nociceptors in rainbow trout can be compared with those in mammals. Differences are evident in rainbow trout nociceptors that do not respond to temperatures below 4 °C due to evolutionary adaptation to lower temperatures (ibid., p. 3). Research clearly shows how A-delta fibers in rainbow trout act in the same way as C fibers in mammals (the African naked mole-rat), responding to various harmful stimuli. C fibers in terrestrial animals contribute to "longer-term pain" (ibid., p. 3), while A-delta fibers signal the first sensation of pain because they conduct stimuli to the central nervous system more quickly. Considering the overall behavior and physiological responses, despite the small number of C fibers, there is a wealth of evidence to confirm that fish feel pain, avoid potentially harmful events, and have the ability to remember.

According to many scientists (Broom 2001; Kavaliers 1998; Sneddon 2003), experiments have shown that fish have the ability to feel and perceive pain: "Thus, life history and ecology can shape the nociception and pain system" (Sneddon 2019, p. 3). Furthermore, a large number of animals learn to associate a painful stimulus with a specific situation and thus can avoid harmful events. Rainbow trout are thought to possess the same direction of transmission of nociceptive information as in mammals, from the peripheral to the central nervous system. Therefore, pain is a by-product of evolution, and all bodily life is moving towards becoming better and more perfect.

Evolutionary mechanisms explain the development of organisms and living beings purely mechanically and naturally, in which physical pain fits perfectly. The principle of natural selection provides a convincing explanation for species changes by adapting living things to certain evolutionary niches, which are continuously evolving over time. Accordingly, biological evolution offers a model for explaining the formation of different organisms over time, which change to adapt to appropriate environmental conditions. Evolutionarily observed pain simply shows how species that were stronger, more resourceful, and more adaptable to the biological environment simply replaced existing species with their persistence for survival and adaptability to the environment and new conditions, their resistance to new diseases, but also producing new diseases that harmed old species. The sedentary lifestyle contributed the most to this; for example, the farm one, which was immediately inhabited by various animals, such as mice and rats that transmitted diseases (Liebermann 2013, pp. 214–18).

What about suffering? Can it be said that suffering is synonymous with pain, that fish suffer as much as humans? "Humans are not merely evolutionary victims of their own genes, but remain responsible for what has gone wrong" (Conradie 2018, p. 8). To

what extent can it then be categorically stated that in this world there is no more room for lived suffering, responsibility for living things and nature, for oneself, since it is somehow imposed that we cannot in any way justify suffering given the existence of pain in the evolutionary body? Insisting on the natural and evolutionary development of pain, however, also implies a reform of our understanding of suffering. In all evolutionary research on pain, not only causal relevance but also the uniqueness of suffering is questioned. The consequence of such an evolutionary understanding of pain results in an introspective lack of effect on distinct human behavior (suffering, use of symbols, and communication), and the causes of such behavior are neither evolutionary nor common with other living beings (Deacon 1997): "However, the emergence of consciousness, self-consciousness, human consciousness and symbolic communication still requires much scholarly interest. There can be no doubt about human distinctiveness (all specimens of all species are indeed distinctive)" (Conradie 2018, p. 5). On the other hand, in our self-understanding and self-perception, suffering is central to our bodily structure. Suffering becomes a meeting place for the perception of everything real and an area of man's compassionate behavior. For "our response to the suffering of the other must be compassion, not an explanation" (Van Hooft 1998, p. 16).

Is then suffering also just a persistent evolutionary "egregious error" (Craig and MacKenzie 2021, p. 7) that is gradually being discovered by exploring pain and the body? The real question is what is meant by suffering? Is suffering interpreted as absolute independence from pain, i.e., the complete independence of our suffering states from the overall physical context? In this case, it is clear that there is no connection between our behavior toward pain and our future and thus our personal identity. Such suffering would be accidental, chaotic, and without identity. Of course, then there would be no connection between our suffering and the activity of the body. However, what if we acknowledge only the psychological conditioning of suffering, and by suffering, we understand only one independence of suffering from physical events and pain? In that case, the notion of suffering would presuppose a complete separation between the physical and mental realms of value. Such separation is contrary to our everyday experience because the existence of psychophysical connection has long been known to human thought. If I eat too much, I will feel nausea (suffering); if I regain lost love, I will be happy (I will stop suffering). Suffering is not just a physically conscious feeling or perception of one mental object of suffering that would be coincidentally related inertly to another separate physical object such as pain. In this way, we cannot derive the meaning of suffering at our own discretion. No criterion could objectively qualify one's suffering condition. Only a person or a living being can suffer. Entering into someone else's suffering probably implies insurmountable epistemological boundaries. As far as suffering is concerned, there are interspaces of interpretation that give preference to the interpretation of suffering as far as the art of living is concerned. Not everyone, but more than one interpretation is logically in-compatible with natural science knowledge. This is also true of the relationship of the natural and spiritual sciences in terms of the relationship between suffering and pain.

*2.3. The Body Is a Unifying Force in Experiencing Pain and Suffering*

There are countless sufferings and pains that we can experience. Suffering and pain are existential experiences that manifest physically (e.g., tissue damage), socially (poverty, violence), existentially (sadness, grief, stress), philosophically (seeking meaning), theologically (absence of God), culturally (racial and gender differences), etc. In this sense, one can speak of the unification of pain and suffering: the body. Pain and suffering, in any case, depending on the body, i.e., on the absence of internal and external reasons that hurt and torment a person. Along with the research of the body, which medicine has been dealing with for centuries, in recent decades, this problem has reached the media space, especially with the development of medical technology. James A. Marcum presents in his work *An introductory philosophy of medicine: humanizing modern medicine* the problem between logos and pathos as follows: "My proposal is that modern medicine must undergo

a revolution not in terms of its logos or ethos but in terms of its pathos. Specifically, pathos can transform the logos of technique and information into wisdom, a wisdom that can discern the best and appropriate way of being and acting for both the patient and the physician. Pathos can also transform the ethos of the biomedical physician's emotionally detached concern or the humane physician's empathic care into a compassionate love that is both tender and unrestricted" (Marcum 2008, p. 14). This quote very much affects the diagnosis of the modern sufferer because "wisdom comes alone through suffering" (Aeschylus 1953, pp. 39–40). Leaving aside, for now, this Marcum's visionary diagnosis of the present position of modern medicine within the social context, the question must be asked how much we still care about the body and what is meant by the body, that human body?

We have already said that a suffering identity is primarily determined by one's bodily life. In the active and passive exposure of one's body to illness and suffering, the physical, moral, social, philosophical, biological, and theological closeness/absence of other people is embodied. The fragile body of another person in illness and suffering requires not only compassionate understanding but also a qualitative response in terms of resolving illness and suffering. Phenomenologists say that it is precisely from the dimension of the body that one must proceed as that which is accessible to sensibility, which is available only as a phenomenon, which as such is presented and ultimately manifests itself as itself (Svenaeus 2014). Everyone has a body, and we all experience it in different ways. The gap between what we would want for our body and what our body is at the moment is shown in the cleavage and impossibility of unambiguous unity, the unity that must suffer the halving on the body and on what thinks that body, the fragmentation of pain and suffering. The body is the center of the human self, or at least what is left of it.

The past of the human body, both theoretical and practical, is very well known to us. The present of the body is biopolitical, bioethical, bio-scientific, biomedical . . . perhaps even biotheodical. The future of the body could be reduced to as little suffering and pain as possible. No matter how many different fragments of the body there were of our body, the body still remains a great unknown, a great mystery, (a sacred drama) practiced by many. A living body allows us everything. It hurts and is at the root of spiritual, social, communicative, and religious suffering. Suffering and pain, in this sense, inevitably refer to the dichotomy or unity of man and God, human and animal, mind and body, healthy and sick. However, the prospects for their solution increasingly rely on the already new and established dualism of a distant and (in)active God and a suffering and sick man.

Our critique of rationalism and ontology does not reject the existence of the living body, nor does it seek to reduce the living body solely to linguistically constructed pain and suffering. On the contrary, pain and suffering as the potential of bodily sensitivity generate an embodied relation, and in that sense, "the body could be said to be a thinking body and to have intentionality prior to the emergence of language and self-consciousness" (Burkitt 1999, p. 75). Western philosophy (and theology as well) was obsessed with an ontological understanding of being in which the subject is an epistemological subject who can understand the world and others rationally and conceptually. Such rationalizing knowledge often turned suffering and pain into manipulative objects. In that suffering and pain, there was too little, if we may say so, flesh. Therefore, we must primarily observe suffering phenomenologically and narratively because suffering and pain can only be retold, experienced, seen, promoted, prevented, etc. Western theology may have reduced pain and suffering too much given the almighty and often (in)active God, pointing out that is the problem of the existence of evil, and that of the moral kind, in general, laid down in man's free will. The physical potential of the body is neglected because the body and bodily feelings are considered inferior material considering that the final answer lies in eschatology: "If the only adequate answer to human suffering is God in the beatific vision, yet God remains incomprehensible and thus mystery in the vision of God, then human suffering is not a problem to be solved" (Miller 2009, p. 846). Our critique is precisely a phenomenological one, one in which the notion of God and understanding God serves as

the highest fundamental way that creates the immanent repression of physical and spiritual pain and suffering. The theological God, who theorizes man, uses concepts to displace pain and suffering too much from the realm of the body. Therefore, our critique holds that the fundamental mode of human existence is a sensory existence that is irreducible to a pure form of life. Man's sensory body, not the conscious mind, is the one that first comes into contact with the world: "In other words, bodily experience is a specific sociocultural event that cannot be extracted from the setting of the experience and ascribed to some universal body" (Rothfield 2005, p. 38). The pain and suffering that belong essentially to our sensibility are not fundamentally irrational, but through them, the conscious subject (man) feels his/her existence.

It has already been said that the Western rational tradition privileged intelligibility over sensibility, considering representation and intelligibility as the first way of perceiving the world. In particular, Husserl prefers the objectifying act over sensibility, the objectification that *turns* our senses into theoretical thought. For Husserl, although sensitivity can offer some sensory elements in the construction of theoretical content, such content does not have a self-sufficient character because sensitivity per se is not intelligible. According to Lévinas, sensory pleasure is a fundamental form of life that precedes reason, representation, and reflection. (Lévinas 2001).

To feel pain and suffering means to be within yourself, to feel your body. In pain and suffering, man separates himself from the other, withdraws into himself. The basic way of life for such a man is dissatisfaction with his sensory needs such as anxiety, loneliness, hopelessness, etc., which contribute to man's agony (Hovey et al. 2017). The misfortune of such a state is manifested in a dissatisfied soul. Pain and suffering are not just about materiality, although at their core is material: the body. The body primarily has a sensory dimension, and the sensory subject can feel the suffering of another person by empathizing with her situation. Suffering is not a reflexive object that can be easily named and understood only by language. It is above all a reasonable appeal and a call to respond qualitatively. Suffering offers itself to our compassion. Therefore, pain and suffering cannot be materialized and informative, but they are essentially existential, individual, communicative. The suffering of another tells the other simply something in common about their life. She is essentially narrative. It is in itself an existential need that requires not only compassion but the alleviation of the existing condition. We cannot treat the suffering of another as objects of knowledge because suffering is infinitely personal and cannot be understood only conceptually. Therefore, understanding and dialogic linguistic relation cannot exhaust the meaning of the relationship between pain and suffering.

## 3. Coping with Suffering and Pain from the Perspective of the Christianity

### 3.1. (Im)passible God

In addition to biological foundations, suffering also has metaphysical roots. Suffering is a very theological (theodicean) issue, perhaps even more so. Humans, as believing beings, cannot reject the challenging speech of God and His action since the sensual experience of God precedes the rational and conceptual understanding of God. Pain, and especially suffering, are manifested in the pre-original *accusation* of God, which not only requires God to engage in salvation but also to be sensitive to human trauma. In other words, it is the affective content, not the representational content, of a person's suffering that would allow God to respond to the person's needs. Theologians say that God's bodily life is already incarnated in human bodily life through his Son. Such incarnation connects the person with the *salvational* (perhaps rather with the eschatological?) action of God. A person's bodily sensibility is the primary relational mode with the world: "The body is neither an obstacle opposed to the soul nor a tomb that imprisons it, but that by which the self is susceptibility itself. Incarnation is an extreme passivity; to be exposed to sickness, suffering, death is to be exposed to compassion, and, as a self, to the gift that costs" (Lévinas 2002, p. 195). The incarnation is not simply an intimate abstract relationship between the person and God but must also refer to a compassionate intracorporeal relationship. A person,

as a passive and powerless being with regard to pain and suffering, is not a coward and indifferent to God's insensitivity, impassibility. Paradoxically, it is courage that makes the person a believer. Can God, therefore, feel a person's suffering?

An online study by Gray and Wegner shows that the more a person suffers, the more she believes. Where does this disproportion between one's own existential state and trust in God come from? According to the research above, "religiosity stems from the dyadic nature of both morality and mind perception" (Gray and Wegner 2010, p. 7). In the mentioned online study, it was shown that the respondents understand the mind mainly in terms of experience (the ability to feel and be conscious) and agency (the ability to do things). In their view, there is a dual type of entity: those who have experience but no action (e.g., babies, dogs, and children), and God, who has action but no experience. In addition, respondents answered that God has an "impoverished mental life" (ibid., p. 7). In moral typecasting theory, moral situations are divided into moral agents (heroes and villains) and moral patients (victims and beneficiaries). According to the dyadic structure of morality, we tend to look for the culprit for bad deeds and a hero whom we will praise and worship. In the absence of rebuke and praise, all the credit for moral acts is *taken* by God as the ultimate moral agent. What is interesting in this study is the fact that although God can do many things, He, unfortunately, remains "incapable of feeling pain, pleasure, or other inner experience" (ibid., p. 14). Human beings with their physical body and physical life seem to have an advantage over God because they have a sensory dimension and experience. Since God has no body and since He has no experience, i.e., *incapable of feeling pain*, He is then impassible and does not suffer (Mullins 2018).

Then why do people need God? According to *cognitivists*, we simply need God as the ultimate moral agent, especially in cases where we do not find a reasonable sequence of events from which suffering arises that cannot be understood and explained: "He [God] is a moral agent but not a moral patient, deserving of our curses and praises but not of our sympathies" (Gray and Wegner 2010, p. 9). Many other researchers such as Boyer feel that people simply unnecessarily attribute moral action to a being they have neither seen nor heard because it is simply a general tendency (Boyer 2001). The reason why people need an extraterrestrial moral agent lies, according to cognitive researchers, in their Hyperactive Agent Detection Device (Barrett 2000). Since humans could not find a responsible moral agent on earth, then all the obscure events were attributed to the supernatural one. Attributing events to an external agent has the benefits of physical and psychological effects because people have a *sense* of control over events. When people do not find a responsible moral agent for major events such as famine or earthquakes, they are simply looking for a far stronger moral agent—God. Especially when it comes to miraculous events, people transfer all the power of such an event to a supernatural being even more (Pargament and Hahn 1986).

What can we say about the above online study? We cannot omit the idea of God as a moral agent, which is one of God's major activities in the universe. This becomes especially clear in *The Transcendental Doctrine of Method*, at the end of *The Critique of Pure Reason*, where Kant writes: "the belief in a God and another world is so interwoven with my moral disposition" (Kant 1998, p. 689) that one cannot exist without the other. Furthermore, we are probably the only creatures who consciously question the necessity and justification of morality, God, and suffering. Research of this type, an online study, said nothing qualitatively new about God's co-suffering, other than showing a large percentage of public opinion about the impassible God. On the contrary, other opinions, albeit theoretical, can be cited, such as that of trinitarian Christian theology, according to which the Father suffered with his Son on the cross (Moltmann 1993): "If God has really participated in a representative sample of human suffering, then God Himself must somehow suffer under the shadow of divine silence"" (Bell 2019, p. 50). However, the suffering of a child lying in a hospital and the suffering of a father who is next to the bed of a sick child is not the same, the suffering of the Son of God and the Father *under the shadow of divine silence*, many will notice. The mentioned child both hurts and suffers; primarily physically. While, in the case

of the father, there is probably only suffering. His body is healthy unlike, for example, a sick child's body. As Lewis puts it, "Whatever fools may say, the body can suffer twenty times more than the mind. The mind has always some power of evasion. At worst, the unbearable thought only comes back and back, but the physical pain can be absolutely continuous" (Lewis 1961, pp. 40–41). Therefore, our notion of suffering necessarily implies a distance from the everyday notion of co-suffering. For the everyday notion of suffering arising from the will and one's own reasons, the co-suffering subject need not risk the body. Suffering with another is one that implies a co-sufferer's responsibility with a certain consciousness and intention and arises primarily from various calculations but not from bodily sacrifice for another. Therefore, the *original (bodily)* suffering is that which rests on bodily pain and not on rational reflection.

Although God cannot suffer in the way that human being suffers, the question remains why did this good and omnipotent God grant us suffering and pain? We could repeat the answer of J. B. Metz, who says that "this question now becomes a major theological question, an absolute eschatological question, a question that can neither be answered nor forgotten, a question for which we, from our side, have no answer; it is the question of 'too much'" (Metz 2006, p. 225). Does a good God then have any control over natural events and human lives? "Or if God has some purpose in mind that is being accomplished, whether directly or indirectly by the occurrence of such catastrophes, does this not prove such a God to be the cruel 'tyrant' that Nietzsche claimed God to be?" (Kropf 2006, p. 183). The question of evil and suffering is ultimately placed in the relationship between man and God, between the experiencer and the experienced, between the unjustifiable and the incoherent, between the salvational God and the *unredeemable* pain . . . Stump's claim that "the problem of suffering is, in a sense, a question about *interpersonal relations*, insofar as the problem has to do with possible morally sufficient reasons for God, an omnipotent, omniscient, perfectly good *person*, to allow human persons to suffer as they do" (Stump 2010, p. 61). The relational and narrative solution is *the most tempting* because only subjective experience of suffering can turn the discourse of pain to one's own physicality which is most concerned by the theodicy of suffering.

### 3.2. Practical Consolation in the Therapeutic Narrative

Narration is very important when contemplating suffering, as human nature is narrative. A man that does not know how to verbalize suffering stays unexpressed, incomplete, and this can result in psychological disease, as stories have therapeutic power. Through secondary experiences, a person can find meaning in suffering and models of solutions. Others, their story and experience, their life and work, give us a broader perspective, facing us toward our internal world and examining it from another point of view.

Suffering is not in vain and without hope. If we look at heroic stories and myths, we see that they share a narrative structure in which the main character voluntarily or forcibly leaves home, an area ruled by order, to step into an area of disorder. The hero's motivation is also important, which plays an important role in whether he will face the set task and whether he will bring something of exceptional value from that task. This rhythm is present in all great world narratives. Only when she escapes from the limitations of *a safe place* when she leaves the conformism of living and enters an unknown area can a person transcend herself and enrich her spirit. Therefore, this approach to literary texts, including the biblical text, can be considered a call to internalize the story and identify with the protagonist who makes Kierkegaard's leap of faith into the unknown. The person is called to move away from the existing in order to realize to some extent the meaning of what surrounds her and affects her as a whole. At this point, we come to *the paradox of Christianity*.

This is best seen in the example of Job who says, "I know that my redeemer lives, and that in the end he will stand on the earth. And after my skin has been destroyed, yet in my flesh I will see God" (Job 19, pp. 25–26). From the initial shock and astonishment at the unjust suffering, Job changes perspective on his condition as he rises from his situation

the moment he faces it. The quoted passage highlights the formulation of the salvation and hope of the suffering person. In the preceding passages, God helps Job reconstruct the distorted state by suffering by examining his self-identity of egoistic righteousness. It was the presence of pain and suffering that was the intersection of his cry for the changeability of his future. Although Job longs for a bodily or intracorporeal relationship, such a relationship is absent. The hope for a change in the sufferer's physical condition was, only temporarily, postponed. Yet Job risked his bodily life by blaming God for his condition. In a figurative sense, Job risked his existence even when the truth he was talking about could be rejected by endangering his faith, his social position, his self: "Suffering recalls Job, as it recalls patients today, to three dimensions of human existence: to a sense of integrity and self; to a recreated relationship with God; and to renewed harmony with the human community" (Fleischer 1999, pp. 480–81). By such actions, Job's place in the aegis of faith is at stake. Job's question of suffering ends in silence, in torment, in hope. It is only on the border of God's suffering and man's cry that the mystical encounter and possible answer to the question of Job and our suffering occurs, to which everyone should answer from their own perspective. In all biblical narrations, as in the one about Job, it is always about searching for exit (exodus), freedom, of looking for a new life path and meaning. The tale of Job is about unbreakable hope that, despite all hardship, suffering will cease at the end and suffering will have a positive outcome. One more positive characteristic of this story is that it has the strength of identification as a healing, therapeutic character. The story of Job is simply a secondary experience that converts into an intimate experience. The listener/reader is identifying with Job, and through Job he finds representational expression of his own dealing with suffering. Through Job, expressing encouragement or conformation of his path highlighting the necessity of change.

Through narrative as a therapeutic (Griffioen 2018) memory of the sufferer and a return to dignity, the person is allowed to pause before the unexpected, unusual response of her God. In the structure of the suffering and tribulation of the innocent (Job), even in her non-existence as denial, there is a God who sees. It is not an effort to construct an indecisive identity of unbearable suffering from God's (in)audible speech, which even God does not pay attention to. We simply cannot respond to the suffering of the world and the community. The meaning of individual suffering is found in the experience and breaking of God's silence and one's own hope in listening to his response. We can poetically exclaim that lamentation is a trace of a *burning bush* that is only experienced up close and that remains inaccessible if one wants to preserve one's own life and not trivialize the mystery of suffering. Therefore, awe is the first attitude in interpersonal relationships, especially in the most sublime relationship, in the love of God where the measure is lost first and foremost and in which it is so easy to slip into selfish presumption and triviality.

### 3.3. God's (In)active Force in Relieving Pain and Suffering

Yet, in the end, the question of suffering concerns primarily meaning, what a person expects from life, and what she seeks at the end of her earthly existence which goes inevitably naturally towards its cessation, towards its extinction: "It is not suffering that destroys people, but suffering without meaning" (Gunderman 2002, p. 42). Therefore, the biblical meaning also indicates that, if a person believes, if she surrenders to that terrible abyss, life is just beginning, and the suffering—physical, moral, mental—stops. Still, the problem remains how to surmount the insurmountable? It also remains unclear how to overcome the infinite distance between almighty God and powerless and suffering man? How does such an endless relationship work and how does it shape the sufferer? How can dynamic intersubjective life function constructively within an infinite relationship?

Possible approaches and solutions are as follow:

(a) It is about a difficult question, how to save God's power, his all-knowingness, and goodness. The Christian answer is this: Christ's suffering and death on the cross marks a foundational historical event but always under God's unexpected merciful approach. Where there was defeat and nothingness, as in the example of Job, God is bringing new

life. Lastly, embodied God becomes a human intermediary to save humanity. In the same way that gods work, this heroic event is not happening outside of the concrete life of an individual in the same way the suffering cannot be abstract but always concrete and personal. Believing that the merciful God exists, with his continuous redemption, truly is immeasurable comfort. In this comfort is rooted the endurance of suffering. It is important to point out Christ's divine–human mediator role, not just his divine mediator but the redeemer-salvation character of his humanity. His humanity represents the character of God's salvation.

(b) Only a rational approach to suffering destroys suffering because of its divided understanding of suffering, and the experience of suffering. Rationalizing suffering is problematic, although necessary, because it views suffering as a static, statistical, documentary, etc., dataset. Experiential and lived suffering requires the transformation of not only the body but also the spirit as it seeks answers. Thus, the religious approach to suffering corresponds to the purification of personal life, implying the renunciation, conversion, modification, and transcendence of existence. The religious treatment of suffering is not intended for the acquisition of knowledge but a person's very existence. Therefore, we give preference to faith over experimental disciplines ones since believers talk about suffering that can very well be in touch with life. Of course, the insights of the experimental sciences are unavoidable when it comes to the pain that manifests in our body. Experimentally observed pain and suffering are devoid of context. They go simply to establish the facts. They worry too little about how to survive today, how to live tomorrow: "The problem of evil and suffering is not a puzzle to be solved but an experience to be lived with" (Dein et al. 2013, p. 200). Since suffering arises from compassion, that is, love for people and self, suffering is no more abstract, it is rather embodied, incarnated, intersubjective. In essence, such incarnated suffering is a well-known concept in Christian theology, which views life as the art of dealing with life's adversities. Suffering as subjective knowledge can serve a person as bliss to the extent that its religious content creates a specific style for the person by transforming her being. Suffering is the wisdom and virtue of life, realized by a certain transformation of the suffering person. Moreover, what is gained in transformation is practical knowledge of life: "In our response to the mystery of suffering, we define ourselves, find our integrity and ultimately shape our ethos" (Fleischer 1999, p. 485).

(c) Taking a human form, he becomes not just a symbol and a sign, but a real embodiment of the divine. In his appearance, it is present the invisible. From this it is then obvious that infinitely merciful God that suffers and dies can also co-suffer.

(d) Constitutively remembering of Christian faith from the beginning can be summarized in the words: "memoria passionis, crucis, mortis et resurrectionis Jesu Christi". Christian remembering of the cross and theology of Christ's death on the cross does not end in defeat but on the experience that good final prevails, that death and suffering are defeated by love. Therefore, Christian handling of suffering does not rest in the feeling of abandonment and weakness but witnesses God's salvation and new life. Initial disappointment transforms into gratitude. Christian remembering on the cross that is not only focused on the pain and suffering but true gratitude of personal salvation happily open to all people. Christ's cross is the synthesis of all suffering.

(e) Therefore, God's plan of salvation is not based on the limits of human possibility. God works, amid human possibility and final man's suffering, in the nothingness that is not removed from radical God's abandonment. Finally, God gives a purpose to every suffering to be allowed and surrendered to, which includes resistance and unrest with suffering for a greater purpose, includes the transformation due to love toward the man. The Christian understanding of pain and suffering does not have the purpose of succumbing to the cross but mainly giving purpose to scuffing through love the same way that Jesus did. The point of following his footsteps is not to be abandoned by God and people or suffer the same consequences; in contrast, as suffering is individual, it remains the challenge that needs to be given meaning.

## 4. Conclusions

It should be said that suffering, however, is not in itself a positive thing. Suffering is properly endured only in a broader sense. Suffering, in fact, is not the meaning of existence. As members of the narrative/story-telling community, sufferers are called upon to be the first to testify that suffering in itself does not have to make sense, but can even capture a person (Strawson 2004, p. 446). On the other hand, the utterance of suffering, its narration, and conversation create a community of people of the same destiny, and suffering is overcome. Pain and suffering are not explicitly categories of science. It concerns all people. Defining pain only through biological, cognitive, medical, and psychological spheres means nod touching the roots and dodging that what is important in Christianity. Pain and suffering cannot stay only on a horizontal plane, only on the level of different sciences. When trying to have an all-encompassing understanding of pain and suffering, there needs to be an inclusion of understanding the divine–human connection.

For this reason, pain must be unbearable in the context of the time (Job knows it; every sufferer knows it, too) because it is measured, temporally observed, it is unbearable, etc. However, if we observe pain and suffering, suffering in the first place, rather than pain, since pain is more related to this physical time, measurable time (hours, days, years, etc.), then it is a burden that should be released as soon as possible. From the perspective of eschatological *time*, what Christianity is talking about, this chronological time is radically changing, and it is easier to bear. That is why pain belongs more to the body and suffering to a person. In the context of our culture that rejects every form of suffering, suffering and pain must be narrated again. Pain and suffering in this context can very well narratively and hermeneutically fit into the category of change, into the category of human growth, and into the category of qualitative growth. That is why history and historical experience, that is, the experience that marked a person's walk with God, which she fondly remembers in moments of misfortune and which carries her, is important for suffering and pain. Although some research shows that a person invokes God more when it comes to bad life circumstances, these past events that person, as a conscious being, can remember, form an integral and central part of her life imbued with pain and suffering. Therefore, an existential experience that a person acquires during her life, and to which she can repeatedly refer, is an unavoidable part of overcoming suffering. In this context, God as the one who acts from the background becomes active through the anamnesis of past temptations. Thus, every pain and suffering is close to God.

**Funding:** This research received no external funding.

**Institutional Review Board Statement:** Not applicable.

**Informed Consent Statement:** Not applicable.

**Data Availability Statement:** Not applicable.

**Conflicts of Interest:** The author declares no conflict of interest.

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
