# Peer review of "(In)active God—Coping with Suffering and Pain from the Perspective of Christianity"

_religions, doi:10.3390/rel12110939_

Round 1
Reviewer 1 Report
- The article is interdisciplinary, but I am concerned that the author does not confuse interdisciplinarity with eclecticism. It would be advisable to add a methodological introduction, specify which scientific disciplines are referred to, define basic terms, and describe the research approach used.
2 I miss the reference to the counterbalance for pain and suffering - so-called well-being, happiness. The so-called Onion Theory of Happiness by Czapiński (2000) could be helpful here.
3) In the thread about God's (in)activity, I would suggest referring to the thought of C.S. Lewis, especially to the book "A Grief Observed".
Author Response
Thank you very much for the review of our manuscript entitled: “ (In)active God – Coping with Suffering and Pain from the Perspective of the Humanities”. We sincerely appreciate all valuable comments and suggestions, which helped us to improve the quality of the article. Our responses to the Reviewer’s comment are described below in a point-to-point manner.
The article is interdisciplinary, but I am concerned that the author does not confuse interdisciplinarity with eclecticism.
The article is interdisciplinary because throughout the paper the author deals with evolutionary, cognitive, medical approach to pain and suffering.
It would be advisable to add a methodological introduction
We agree with Reviewer that a methodological introduction should be added.
specify which scientific disciplines are referred to
„We, therefore, consider it important to answer the question of suffering from a humanistic perspective, since suffering is diverse and cannot be measured empirically.“
define basic terms:
The concept of pain and suffering is defined in our chapter The mystery of pain and suffering.
and describe the research approach used.
I agree with Reviewer that the research approach is insufficiently described.
2 I miss the reference to the counterbalance for pain and suffering - so-called well-being, happiness. The so-called Onion Theory of Happiness by Czapiński (2000) could be helpful here.
The counterbalance of pain and suffering is placed in the relationship between a God's incomprehensibleness and a man who seeks a firm and sure answer. This is the fundamental question of how to cope with pain, and especially suffering. The different religions provide different answers to the question of pain and suffering. However, these answers are not final. Unfortunately, we can only talk about suffering. Suffering cannot be solved once and for all.
We would like to thank the Reviewer for the article by Czapiński.
3) In the thread about God's (in)activity, I would suggest referring to the thought of C.S. Lewis, especially to the book "A Grief Observed".
I agree with Reviewer that the chapter on God’s (in)action can be reworked.
Reviewer 2 Report
-A very interesting point of view, even if the topic of this paper is not original or new - since it addresses a subject debated from IV century, that of "Theopaschite Doctrine", The paper's details make the reading interesting for the angle it seeks to re-approach this ancient theme. -anyways, the two theological consequences your assertion draws might endanger the beauty of your manuscript, them being: 1. if God-Man willingly sensitized his body to feel suffering at the highest levels as required by the sacrifice of One for many, that 'meta-suffering' [and that pain and suffering go near to suicide for the awareness and self-infliction] or 2. if God-Man desensitized/trained his body not to feel suffering at all [thus He was not at all engaged in so-called ' Christ passion']. However, further debates and axiological colloquial explanations are needed to conclude this theme. -Overall, it is an excellent paper I've enjoyed reading with very good built argumentation!Author Response
We would like to thank Reviewer for taking the time and effort necessary to review the manuscript. We sincerely appreciate all valuable comments and suggestions, which helped us to improve the quality of the manuscript.
„The paper's details make the reading interesting for the angle it seeks to re-approach this ancient theme. -anyways, the two theological consequences your assertion draws might endanger the beauty of your manuscript, them being“
Thank you for pointing this out. We agree with this comment. Therefore, the chapter on God’s (in)action will be reworked.
Reviewer 3 Report
This is readable, though the use of quotation marks is odd and not the sort seen in English language punctuation marks.
It is not at all clear what the author or authors intend to do with this piece. The second last paragraph begins "If we have managed to interest the reader in the old and difficult question of pain, suffering, and God...then our effort succeeded. If this is the criterion for the success of the work, it simply does not belong in a scholarly journal. Surely a scholarly journal should publish papers that do more than merely seek to interest a reader!
The title of the article mentions "(In)active God." Presumably this would have included engaging Caputo's "weak theology" or Oord's God Can't or other sources and conversations, even the so-called "Death of God" theology of many decades ago. Yet none of this was done.
The is a section on "the body" but nothing really on embodiment or related contemporary theological moves, nor similar ones in the field of religious studies. Some of the citations are interesting yet it is unclear why these were chosen and they are certainly insufficient for the discussion. The literature is simply much bigger. The swing from a few cherry-picked authors and cherry-picked scriptures is neither convincing nor complete.
At other points of the text, claims are made ("an online study shows...") but no citations or further discussion is given. What is the main point? How does this work cohere?
This would have been a somewhat interesting paper if various strands of the literature had been engaged in some definitive way. The authors discuss something they call "narratization," or example. It is unclear why the author or authors are inventing this word.
Every paragraph seemingly raises more questions than are answered. The flow of the core claims and overall argument is not at al clear. Nonetheless, the core topics are important and should be addressed in a novel way. I was hoping that this article would do that. I can imagine that the author or authors might overhaul this work with a great deal more clarity of their intent, use of sources, and core claims. However, the author or authors might be better served by starting from scratch.
Author Response
Thank you very much for the review of our manuscript entitled: “ (In)active God – Coping with Suffering and Pain from the Perspective of the Humanities”. We sincerely appreciate your comments. We’re sorry to read you had a frustrating experience, but we really appreciate you bringing this issue to our attention.
This is readable, though the use of quotation marks is odd and not the sort seen in English language punctuation marks.
The article has been prepared in accordance with the journal template, the layout guide and a published paper.
It is not at all clear what the author or authors intend to do with this piece.
That pain and suffering are one problematic, the controversial and complex phenomenon has already been indicated, and will be explained in the rest of the paper.
The second last paragraph begins "If we have managed to interest the reader in the old and difficult question of pain, suffering, and God...then our effort succeeded. If this is the criterion for the success of the work, it simply does not belong in a scholarly journal. Surely a scholarly journal should publish papers that do more than merely seek to interest a reader!
We absolutely agree with you. That essayistic sentence is pointless and unnecessary.
The title of the article mentions "(In)active God." Presumably this would have included engaging Caputo's "weak theology" or Oord's God Can't or other sources and conversations, even the so-called "Death of God" theology of many decades ago. Yet none of this was done.
I only agree with you here.
The is a section on "the body" but nothing really on embodiment or related contemporary theological moves, nor similar ones in the field of religious studies. Some of the citations are interesting yet it is unclear why these were chosen and they are certainly insufficient for the discussion. The literature is simply much bigger. The swing from a few cherry-picked authors and cherry-picked scriptures is neither convincing nor complete.
Suffering is primarily determined by one’s bodily life. In the active and passive exposure of one's body to illness and suffering, the physical, moral, social, philosophical, biological… theological closeness/absence of other people is embodied. The fragile body of another person in illness and suffering requires not only compassionate understanding but also a qualitative response in terms of resolving illness and suffering. Phenomenologists say that it is precisely from the dimension of the body that one must proceed as that which is accessible to sensibility, which is available only as a phenomenon, which as such is presented, and ultimately manifests itself as itself (Svenauas 2014). Everyone has a body and we all experience it in different ways. The gap between what we would want for our body and what our body is at the moment is shown in the cleavage and impossibility of unambiguous unity, the unity that must suffer the halving on the body and on what thinks that body, the fragmentation of pain and suffering. The body is the center of the human self, or at least what is left of it.
At other points of the text, claims are made ("an online study shows...") but no citations or further discussion is given. What is the main point? How does this work cohere?
Our comments rely upon „An online study by Gray and Wegner shows that the more a person suffers, the more she believes. Where does this disproportion between one’s own existential state and trust in God come from? According to the research above, “religiosity stems from the dyadic nature of both morality and mind perception“ (Gray and Wegner 2010, p. 7). In the mentioned online study, it was shown that the respondents understand the mind mainly in terms of experience (the ability to feel and be conscious) and agency (the ability to do things).“
This would have been a somewhat interesting paper if various strands of the literature had been engaged in some definitive way. The authors discuss something they call "narratization," or example. It is unclear why the author or authors are inventing this word.
The word „narratization“ is mentioned only once in the article.
Every paragraph seemingly raises more questions than are answered.
This is true. We can only repeat: The question is more important than the answer.
The flow of the core claims and overall argument is not at al clear. Nonetheless, the core topics are important and should be addressed in a novel way. I was hoping that this article would do that. I can imagine that the author or authors might overhaul this work with a great deal more clarity of their intent, use of sources, and core claims.
However, the author or authors might be better served by starting from scratch.
Unfortunately, starting from scratch might not be a good idea. The article definitely must be improved.
Round 2
Reviewer 3 Report
The work is sufficiently improved.